# Stevia (*Stevia rebaudiana*) Improves Carotenoid Content in Eggs When Fed to Laying Hens

**DOI:** 10.3390/foods11101418

**Published:** 2022-05-13

**Authors:** Vasil Radoslavov Pirgozliev, Isobel Margaret Whiting, Kristina Kljak, Stephen Charles Mansbridge, Atanas Georgiev Atanasov, Stephen Paul Rose, Stanimir Bojidarov Enchev

**Affiliations:** 1National Institute of Poultry Husbandry, Harper Adams University, Shropshire TF10 8NB, UK; iwhiting@harper-adams.ac.uk (I.M.W.); smansbridge@harper-adams.ac.uk (S.C.M.); sprose@harper-adams.ac.uk (S.P.R.); 2Faculty of Agriculture, University of Zagreb, 10000 Zagreb, Croatia; kkljak@agr.hr; 3Ludwig Boltzmann Institute for Digital Health and Patient Safety, Medical University of Vienna, Spitalgasse 23, 1090 Vienna, Austria; atanas.atanasov@dhps.lbg.ac.at; 4Institute of Genetics and Animal Biotechnology of the Polish Academy of Sciences, 05-552 Magdalenka, Poland; 5Agricultural Institute, 9700 Shumen, Bulgaria; stanimir_en@abv.bg

**Keywords:** stevia, phytogenic, carotenoids, eggs, eye health, human health, laying hens

## Abstract

*Stevia rebaudiana Bertoni* is a shrub with leaves that have a high concentration of carotenoids such as lutein and zeaxanthin. Egg yolks are a bioavailable source of lutein and zeaxanthin. The consumption of these carotenoids has been linked with improved human health. To investigate the impact of dried stevia leaves at 0%, 1% and 2% on the quality variables, the chemical composition and antioxidant content of eggs, the experiment involved 90 Hy-Line Brown laying hens, housed in 30 enriched layer cages, in groups of three from 22 to 26 weeks of age. The impact on the internal qualities of stored eggs was also examined. Yolks from hens fed stevia had an enriched color compared with the controls. At the end of the experiment, the whole egg, without shell, of birds fed 2% stevia had a higher total carotenoid content (*p* < 0.001) compared with birds fed 1% and 0% stevia, i.e., 5.16 (µg/g), 4.23 (µg/g) and 2.96 (µg/g), respectively. Storage reduced albumen height and increased albumen pH (*p* < 0.001). Stevia supplementation did not interact (*p* > 0.05) with storage time among the egg quality variables. Consuming eggs from hens fed stevia may increase carotenoids in human diet.

## 1. Introduction

The application of dietary plant extracts to poultry feeds is gaining recent interest, particularly for their use as an alternative to antimicrobial growth promotors [1,2,3]. Plant extracts, also known as phytogenics, phytobiotics or botanicals, are products derived from plant origin and refer to a variety of ingredients including herbs, spices, essential oils, and oleoresins [4]. Dietary phytogenics have been found to influence nutrient availability, growth performance, feed intake, endogenous secretions, and immune response [1,5]. They have also been found to possess antioxidative properties [5,6,7,8].

Stevia (*Stevia rebaudiana, Bertoni; STE*) is a small, leafy, perennial shrub belonging to the *Asteraceae* family. It is native to parts of South America, where, historically, it has been used medicinally by indigenous peoples. [9]. Stevia is also well known for its use as a natural sweetener for human consumption. Steviol glycosides, such as stevioside and rebaudioside, extracted from the leaves of stevia, have been found to be approximately 300 times sweeter than sucrose [10,11]. In vitro work showed that stevioside, carotenoids, phenolic compounds and flavonoids in STE might also be involved in antioxidant defense mechanisms to help relieve stress [12,13]. Research by Pirgozliev et al. [14,15] showed that the lutein and zeaxanthin from STE can be accumulated in the liver of chickens, thus improving the antioxidative status of birds through enhancing dietary antioxidant availability and potentially increasing resistance to diseases. Lutein and zeaxanthin belong to the class of oxygen-containing carotenoids known as xanthophylls and can be found abundantly in green leafy vegetables, colorful fruit, maize, pepper, various animal tissues and egg yolk [16]. An increase in the dietary intake of lutein and zeaxanthin in humans is associated with improvements in visual function [17]. A daily intake of 5–10 mg/day of lutein/zeaxanthin can reduce the risk of age-related macular degeneration (AMD) [17]. The daily consumption of those xanthophylls in developed countries, such as the USA, is relatively low, e.g., 1–2 mg/day, thus increasing the risk of reduced human macular pigment optical density [17,18]. In the EU alone, the annual costs associated with managing the effects of severe AMD is approximately EUR 89.46 billion [19].

Eggs are an essential part of the human diet. Globally, the average consumption of eggs each year per capita ranges from about 62 in India to in excess of 358 in Mexico, as the average consumption is about 161 eggs per person per year [20]. Ortiz et al. [21] reported that the total xanthophyll content in the egg yolk of hens fed yellow or orange maize based diets, or free range hens may exceed 20 µg/g egg yolk. Comparable with other carotenoids, the bioavailability of lutein can be enhanced when consumed along with lipids [22]. Bioavailability can also be influenced depending on the food source and matrix. For example, lutein derived from lutein-enriched eggs provides more bioavailable lutein than supplements or spinach [23].

Stevia exhibits natural variation in carotenoids that are associated with visual function and overall human health benefits [17]. These include lutein, zeaxanthin, β-cryptoxanthin, and β-carotene, as well as vitamin E and coenzyme Q10 [14,15]. The idea of the biofortification of hen’s eggs has been well applied [21,24,25,26], although information on the impact of feeding STE to laying hens on egg carotenoids is lacking.

Information on potential changes to the physical characteristics of stored eggs when feeding hens diets containing STE is also very limited [27]. Niemiec et al. [28] reported that the dietary addition of antioxidants did not affect egg weight after 20 days of storage at 12 °C, although Omri et al. [27] found that feeding antioxidants reduced the losses of egg, albumen and the shell of eggs stored at 4 °C for 30 days. Whiting et al. [29] also found no effect of dietary antioxidants on stored egg quality variables. In general, further information is required to determine the effect of dietary STE on the quality variables, chemical composition and antioxidant content of both fresh and stored hen eggs.

The aims of this study were to (1) establish whether increasing dietary levels of STE in hen feeds can increase the concentration of lutein, zeaxanthin and total carotenoids in hen eggs, (2) to examine the effects on egg and eggshell quality variables, and (3) to investigate the impact of dietary STE on the quality variables of stored eggs. It has been hypothesized that feeding STE to laying hens will increase carotenoid content in eggs without having an impact on egg production variables.

## 2. Materials and Methods

This experiment was conducted at the National Institute of Poultry Husbandry (NIPH) and approved (0150-201202-STAFF) by the Research Ethics Committee of Harper Adams University (UK). This manuscript complies with the ARRIVE guidelines [30].

### 2.1. Study Design, Laying Hen Management and Egg Collection

#### 2.1.1. Dietary Formulation

From 20 weeks of age the birds were fed a basal diet formulated to meet the nutrient requirement of the hens, containing 11.56 MJ/kg AME and 172 g/kg crude protein (Table 1). The experiment started when birds were 22 weeks old. The basal diet was then split in three parts. One part was fed as it was and is named Control (C). The second diet was made by adding 10 g of milled dry stevia leaves to 1000 g of the C diet (1% STE). A third diet was mixed by adding 20 g of milled dry stevia leaves to 1000 g of the C diet (2% STE). Dietary STE was from the cultivar Stela [31] and was produced and supplied by the Agricultural Institute, Shumen, Bulgaria. The experimental diets were fed in a meal form for four weeks, between 22 and 26 weeks of age and did not contain any coccidiostat, antimicrobial growth promoters, prophylactic or other similar additives.

The vitamin and mineral premix contained vitamins and trace elements to meet the requirements specified by the breeder. The premix provided (units/kg diet) the following: ^1^ Premix (per kg feed): Vit A (retinyl acetate) 10,000 IE; Vit D3 (cholecalciferol) 2000 IE; Vit E (dl-a-tocopherol) 25 mg; Vit K3 (menadione) 1.5 mg; Vit B1 (thiamin) 1.0 mg; Vit B2 (riboflavin) 3.5 mg; Vit B6 (pyridoxine-HCl) 1.0 mg; Vit B12 (cyanocobalamin) 15 µg; niacin 30 mg; D-pantothenic acid 12 mg; choline chloride 350 mg; folic acid 0.8 mg; biotin 0.1 mg; iron 50 mg; copper 10 mg; manganese 60 mg; zinc 54 mg; iodine 0.7 mg; selenium 0.1 mg.

#### 2.1.2. Experimental Design

A total of 90 Hy-Line Brown laying hens were housed in 30 enriched layer cages (Hellmann Poultry GmbH & Co. KG, Vechta, Germany), in groups of three from 22 to 26 weeks of age. The experiment was conducted using a randomized block design as each diet was fed to 10 cages following randomization. The rearing conditions followed industry recommendations (www.hyline.com, accessed 1 April 2022). The birds had *ad libitum* access to feed and water.

#### 2.1.3. Hen Performance, Egg Production and Determination of Egg Quality

The hens were individually weighed at the beginning of the study, at 22 weeks of age. Feed intake (FI) of each cage was recorded and presented on a per cage basis for the study period. Egg numbers were recorded daily and egg weight was determined once per week, assuming that this is the average egg weight for the week. Feed conversion ratio (FCR) for egg production and egg production (%) were determined as previously described [32,33].

Egg and egg shell quality analyses were performed on a total of 30 eggs which had been collected, one egg from each cage, on the last day of the experiment (26 weeks old). The analyses of the eggs were completed after one day of storage at 15 °C. Eggs were individually weighed. Albumen height (AH) and Haugh units (HU) were measured using Technical Services and Supplies (TSS) Egg Ware (Chessingham Park, Dunnington, York, YO19 5SE, England) as previously described [32,33]. Yolk color was measured using DSM YolkFanTM (YF). The yolk and egg were then separated to determine the pH of each using a FC2133 Foodcare pH and temperature electrode probe (Hanna Instruments Ltd., Leighton Buzzard, UK). Eggshells were washed and left to dry for 24 h in an air-forced oven at 40 °C with the membrane in place. Once dried, eggshells were weighed and shell thickness was measured by averaging the measurements taken at three locations on the egg’s equator using a TSS QCT shell thickness micrometer.

At the end of the study, one egg from each cage was obtained to determine the impact of the dietary treatments on shell color using a Konica Minolta Chroma Meter CR-400/410 (Minolta, Tokyo, Japan). Analysis followed the Commission Internationale de L’Eclairag (CIE) color system, where L* indicates lightness, while a* and b* are chromaticity coordinates. The contents of each egg were then broken out, freeze dried and ground using a mortar and pestle. The impact of the dietary treatments on the color of the internal contents was determined by Konica Minolta Chroma Meter as previously described.

#### 2.1.4. Proximate Analysis of Experimental Diets and Eggs

At the end of the study, another egg was collected from each cage. The eggshell was removed, ground and used for mineral analysis. The yolk and albumen were freeze dried, ground and analyzed for their chemical composition. Dry matter (DM) of the feed samples, yolk and albumen were determined by drying samples in a forced draft oven at 105 °C until a constant weight [34]. Crude protein (6.25 × N) in samples was determined by the combustion method [35] using a LECO FP-528 N (Leco Corp., St. Joseph, MI, USA). Fat (as ether extract) in samples was extracted with diethyl ether by the ether extraction method [36] using a Soxtec system (Foss Ltd., Warrington, UK). Ash content of the eggshells were determined by pre-ashing using a Bunsen burner and placing samples in a muffle furnace at 550 °C for 6 h. Mineral concentrations in the diets and eggshells were determined by inductively coupled plasma emission spectrometry (Optima 4300 DV Dual View ICPOE spectrometer, PerkinElmer, Beaconsfield, UK), as described by Tanner et al. [37].

One egg from each treatment was collected on days 0, 7, 14, 21 and 28 of the experimental period. The eggs were broken, eggshell was removed and the content of yolk and albumen was freeze dried, milled and used to determine the carotenoids concentration throughout the duration of the study. The concentration of carotenoids and total vitamin E in the feed samples, yolk and albumen were determined as previously described [6,38,39,40].

#### 2.1.5. Egg Storage Investigation

At the end of the final week of the study (at 26 weeks of age), three eggs were collected from each cage and stored for 28 days at 15 °C. Egg quality measurements were taken every two weeks (0, 2 and 4 weeks following storage). One egg from each cage was tested at each time period to determine studied values. Measurements determined included albumen height, HU, albumen and yolk pH and yolk color values. The same egg was used to record egg weight over time.

#### 2.1.6. Statistical Analysis

Egg data were analyzed using Genstat (18th edition) statistical software package (IACR Rothamstead, Hertfordshire, UK). Comparisons among egg quality, proximate analysis and minerals in eggs were performed by ANOVA. Comparisons between the carotenoid content of the eggs were carried out by employing a two-way ANOVA using a 3 × 5 factorial design (dietary stevia levels × time of egg collection). Comparisons among the studied variables for the storage investigation were performed by a two-way ANOVA using a 3 × 3 factorial design (dietary stevia levels × storage period). All tests were considered significant at *p* < 0.05. Data are expressed as means and their pooled standard errors (SEM).

## 3. Results

### 3.1. Effect of Dietary Stevia on Egg Hen Performance, Production, Egg Quality, Proximate and Mineral Analysis of Eggs at Study End Point

Dietary stevia had a much higher concentration of carotenoids, primarily lutein, compared to the basal diet (Table 2). However, the content of total vitamin E was very similar in the stevia and basal diet. The experiment started when the birds were 22 weeks old and their average body weight was 1.752 kg (SD = ±0.0723).

The effects of dietary stevia on hen performance and egg quality are shown in Table 3. There were no differences (*p* > 0.05) in feed intake, average egg weight and FCR for egg production due to the experimental diets. The only difference observed was in egg production (%), as birds fed 1% STE laid eggs as those fed C (*p* > 0.005), but less than birds fed 2% STE (*p* = 0.045). The effect of dietary stevia on both internal and external egg quality variables at the endpoint shows there was no significant difference across the three treatment groups for egg weight, HU, albumen height or yolk and albumen pH (*p* > 0.05). No differences were observed across the three treatment groups for L* (lightness) or a* (redness) for yolk color (*p* > 0.05). However, an increase (*p* = 0.027) in b* (yellowness) can be observed for the inclusion of dietary stevia at 2%. The response to b* was supported by an increase in the YF scores for both 1% and 2% STE-containing diets compared to the control diet (*p* < 0.001). No effects were observed for eggshell color measurements (*p* > 0.05). A proximate analysis of the yolk and albumen and mineral analysis of the eggshell are shown in Table 4. There was no effect (*p* > 0.05) of dietary stevia on dry matter, ash, crude protein, fat or mineral content of the eggs.

### 3.2. Effect of Dietary Stevia on Carotenoids and Vitamin E Content of Yolk and Albumen at Different Time Points

There was a diet by time interaction (*p* < 0.05) for lutein, zeaxanthin and total carotenoids concentration in eggs (Table 5). Compared to day 0, the concentration of all carotenoids increased by day 7, independent of diet being fed. However, the increase was higher (*p* < 0.001) in eggs from birds fed stevia compared to the control. At day 14, there was a decrease in all antioxidants compared with day 7, although birds fed stevia had a higher concentration of lutein and total carotenoids compared to control. At days 21 and 28, the concentration of lutein, zeaxanthin and total carotenoids was higher (*p* < 0.001) in birds fed the C + 1% STE compared with C, while birds fed the C + 2% STE had an even higher concentration compared with both C and C + 1% STE. The experimental diets did not have an impact on the vitamin E concentration of the eggs (*p* > 0.05). However, there were differences for egg vitamin E concentration between the different time periods (*p* < 0.001). The data in Table 5 suggest that feeding stevia for a longer period increases the carotenoid content in the eggs, although the results on vitamin E content were inconsistent (*p* < 0.001).

### 3.3. Effect of Dietary Stevia and Length of Storage on Egg Quality Variables

The impact of dietary STE on the quality of stored eggs is shown in Table 6. The variables measured are inconsistent. Feeding C + 1% STE increased yolk pH compared to the control (*p* = 0.027), but there was no difference (*p* > 0.05) in pH between the control and C + 2% STE diets. However, feeding C + 1% STE increased yolk color intensity compared with the control while feeding C + 2% STE increased yolk color intensity compared to both the control and C + 1% STE diets. Dietary STE did not have a significant impact (*p* > 0.05) on the egg weight, AH, HU and albumen pH of the stored eggs. Compared to the control, egg weight decreased after 28 d of storage (*p* = 0.40), but egg weight at 14 d of storage did not differ (*p* > 0.05) from the control or 28 d storage. Over the 28 d storage period, AH and HU decreased over time (*p* < 0.001). Yolk pH and yolk color decreased during the first two weeks of storage (*p* < 0.001), but their values reached the pre-storage levels after four weeks of storage. There were no interactions observed (*p* > 0.05).

## 4. Discussion

This study aimed to investigate the impact of dietary stevia on yolk pigmentation, and carotenoid concentration when fed to Hy Line Brown laying hens at an early stage of production. Feed intake, egg chemical composition and egg production were also measured. The observed small differences between dietary calculated and determined composition is most likely due to the differences between the composition of the ingredients that were used in the present study and the values in the software used for dietary formulation for the same ingredients. Given the length of the study and the number of replications, it is not expected that this may have an impact on the study outcome. The production and egg quality results were within the expected range for eggs from Hy Line Brown laying hens at this stage of production. The lack of differences between the control and 2% STE regarding production variables suggest that STE can be incorporated in laying hen diets at 2%. However, the determination of an optimum inclusion level for both, antioxidant content in eggs and production variables requires more research.

The study compared the yolk pigmentation and carotenoid deposition of eggs produced from diets with the same proximate composition but with different dry stevia leaf inclusion. Yolk color has been previously found to reflect the content of carotenoids in hen feed, while yolk color has an aesthetic effect on consumer preference [21,41,42]. The diets containing stevia resulted in egg yolks with a higher YF color and b* index values (indicating yellowness) compared with the control diet. Moreno et al. [42] and Ortiz et al. [21] obtained higher results in YF yolk color when feeding yellow or orange maize compared to our results. This may be explained by the high inclusion rate of maize in the diets (over 500 g/kg) compared to stevia (20 g/kg), although the stevia contained 55.8 µg/g carotenoids compared to 5.7 µg/g and 24.9 µg/g for yellow and orange maize, respectively [21].

Although β-carotene and β-cryptoxanthin were present in the stevia leaves, they were not detected or found in negligible amounts in the eggs. When carotenoid compounds such as β-carotene and β-cryptoxanthin are converted into vitamin A, they lose their pigmenting properties which may have influenced this fraction in the yolk [39,43]. In the present study, the obtained range of total carotene fraction in the yolk is comparable to values reported in previous studies, irrespective of the dietary content or source [21,25,39]. Kljak et al. [40] also reported an initial increase in yolk carotenoids in a study, followed by a dip and further increase. Weekly fluctuations occurred among individual carotenoids, markedly so β-cryptoxanthin and β-cryptoxanthin compared with lutein and zeaxanthin. Despite the deposition of xanthophylls into egg yolk being a relatively quick process, it could take at least attain a stable response to dietary pigments [40,44,45], thus supporting the results obtained in our study.

Lutein, zeaxanthin, and meso-zeaxanthin are the macular pigment carotenoids [16]. The yellow color of the central retina (*Macula lutea*) in human eye is due to the presence of the carotenoid pigments lutein and zeaxanthin [18]. The high serum level of macular carotenoids is suggested to play a role in the protection of the retina against light-induced damage [46]. However, the macular carotenoids cannot be synthesized *de novo* and must be obtained through diet [16]. The macular carotenoid concentration in yolks produced with the C + 2% STE diet was higher than in yolks produced with the C + 1% STE diets and the control in the reported study. It must be noted, however, that in the reported study, carotenoids were determined in the mix of yolk and albumen together, despite carotenoids being found in the yolk only. Since the yolk is about one-third of the egg without shell [47] the actual carotenoid content in yolk is over three times higher than reported in the whole egg in this study. This means that the overall carotenoid content in the yolk at the end of the study for hens fed the control, C + 1% STE and C + 2% STE diets are about 9 µg/g, 14 µg/g and 17 µg/g, respectively. Thus, in the birds fed stevia, the carotenoid level was higher than the levels in eggs produced by hens fed white maize, but comparable to the eggs of free range, herb fed, colored maize or commercially available marigold fed birds [21,25,39]. This also suggests that planting stevia shrubs in the areas of free-range reared hens may increase the carotenoid content in eggs. Thus, suggesting that further research on the impact of higher dietary stevia inclusion on egg carotenoid content is warranted.

Dietary STE had about 1.1 µg/kg or 6.2% less vitamin E compared to the control diet, thus suggesting an explanation for the higher vitamin E concentration in eggs laid at the beginning of the study, compared with the following time periods. Although numerically, only birds offered STE consumed less feed, which may be connected to inconsistent egg vitamin E concentration. Although diet is more influential than heredity in its effect on variability of egg composition, the content of protein and fat is directly influenced by birds’ genetics [47], thus supporting the observed lack of differences in chemical composition of eggs in this study.

As expected, albumen height decreased with the length of storage while albumen pH increased [48]. The reduced egg weight at the end of the four-week storage period results from water exchange between yolk and the egg white and from water and carbon dioxide loss through the eggshell pores [20]. Egg yolks are known for their high fat content and are therefore susceptible to lipid oxidation. In this study, hens were able to deposit antioxidants in the form of carotenoids from diets into their egg yolks that could protect the lipids during egg processing. However, stevia addition in laying hen diets does not affect the impact of storage on egg weight or the internal quality of eggs produced. Usually, eggs are stored at room temperature and are considered as “fresh” up to 28 days after laying [20]. In addition, research by Nimalaratne et al. [49] has demonstrated that the antioxidant activity of egg yolk was globally unchanged during six weeks of retail storage. Thus, suggesting that the relatively short storage period and the controlled environment, i.e., 15 °C, may have contributed to the lack of interaction between dietary stevia and length of storage on the studied egg quality variables in this study. In addition, diets were supplemented with synthetic vitamin E (slightly exceeding NRC, 1994 recommendation of 12 IU/kg), thus providing enough for hens to deposit antioxidants from diets into their egg yolks that could protect the lipids during egg storage. This may further explain why STE addition in the laying hen diets did not affect the quality of stored eggs.

## 5. Conclusions

The supplementation of hen diets with stevia increased the carotenoid content and yellowness of the yolks compared to the control diet. The C + 2% STE diet resulted in higher carotenoid content in eggs compared to the control diet and the C + 1% STE diet (17 µg/g vs. 9 µg/g and 14 µg/g, respectively, expressed as in yolk only). The carotenoid concentration in the eggs of birds fed stevia was comparable to free range, colored maize or commercially available marigold fed hens, as reported in the literature. Stevia addition in laying hen diets did not affect the impact of storage on the internal quality of stored eggs in this study. Increasing the dietary sources of lutein and zeaxanthin for humans could be achieved by including stevia leaves in feeds for laying hens to increase the concentration of macular carotenoids in egg yolks. Further research into the suitability of using stevia in poultry feeds should be carried out to determine its economic relevance.

## Figures and Tables

**Table 1 foods-11-01418-t001:** Formulation of the experimental basal diet.

Ingredients (%)	Basal Diet
Barley	10.00
Wheat	53.50
Soya meal	17.50
Full fat soya	5.00
L Lysine	0.05
DL Methionine	0.15
L Threonine	
Soya oil	2.00
Limestone	10.00
Monocalcium Phosphate	0.80
Salt	0.25
Sodium bicarbonate	0.15
Layer Vit-Min Premix ^1^	0.10
Titanium Dioxide	0.50
Calculated provisions	
AME (MJ/kg)	11.56
CP (g/kg)	172.0
Oil (g/kg)	43.0
Av Lysine (g/kg)	8.26
Meth + Cysteine (g/kg)	0.664
Ca (g/kg)	41.68
Av P (g/kg)	3.08
Determined values	
DM (g/kg)	905
GE (MJ/kg)	14.63
CP (g/kg)	167
Oil (g/kg)	47.5
Ca (g/kg)	43.2
P (g/kg)	5.0

**Table 2 foods-11-01418-t002:** Antioxidant composition of stevia and the basal diet (analyses were performed in duplicates).

Determined Values (µg/g)	Stevia	Basal Diet
Lutein	50.4	0.514
Zeaxanthin	4.1	0.061
β-cryptoxanthin	0.1	nd
β-carotene	1.3	0.052
Total carotenoids	55.8	0.627
α-tocopherol	11.9	2.095
γ-tocopherol	3.0	11.930
δ-tocopherol	1.3	3.223
Q10	18.0	1.204
Vitamin E	16.186	17.248

nd = not determined.

**Table 3 foods-11-01418-t003:** Effect of dietary stevia (STE) on hen performance and egg quality variables at study endpoint.

	C	C + 1% STE	C + 2% STE	SEM	*p* Value
Feed intake (g/bird/day)	118	113	114	2.3	0.338
Egg mass (g/bird/day)	54.8	51.1	54.0	1.95	0.392
FCR egg production (g:g)	2.160	2.226	2.136	0.0858	0.748
Egg production (%)	93.5 ^ab^	90.6 ^b^	94.7 ^a^	1.064	0.045
Egg weight (g)	58.2	56.6	56.2	2.00	0.750
Albumen height (mm)	8.44	8.27	7.88	0.429	0.646
Haugh unit	91.7	91.4	89.0	2.42	0.693
Albumen pH	8.43	8.32	8.46	0.053	0.187
Yolk pH	6.28	6.38	6.23	0.051	0.133
Eggshell thickness (mm)	0.333	0.338	0.356	0.0135	0.473
Eggshell weight (g)	5.20	5.02	5.50	0.272	0.470
Yolk color (DSM YolkFan^TM^)	2.30 ^a^	3.39 ^b^	3.60 ^b^	0.286	<0.001
Internal egg color (Minolta Chroma Meter)					
L*	81.6	80.3	80.7	2.36	0.919
a*	0.43	−0.18	−0.11	0.520	0.664
b*	17.49 ^a^	19.81 ^ab^	21.07 ^b^	0.854	0.027
Eggshell color (Minolta Chroma Meter)					
L*	57.86	57.74	57.87	0.777	0.991
a*	20.50	20.13	20.39	0.531	0.885
b*	32.50	31.49	32.13	0.536	0.424

C = control diet; SEM = pooled standard error of means; *p* = Fisher’s probability; FCR = feed conversion ratio; L* = lightness; a* = redness; b* = yellowness; values in a row with different letters differ significantly.

**Table 4 foods-11-01418-t004:** Proximate analysis of the yolk and albumen and mineral analysis of the eggshell after feeding dried stevia leaves to laying hens for four weeks.

	C	C + 1% STE	C + 2% STE	SEM	*p* Value
Yolk and albumen					
Dry matter (g/kg)	302.5	297.5	293.1	4.45	0.344
Crude protein (g/kg)	374.8	374.8	379.0	7.19	0.898
Crude fat (g/kg)	211.6	224.4	225.5	6.18	0.238
Egg shell					
Ca (g/kg)	82.5	79.5	81.3	2.38	0.674
P (g/kg)	4.7	5.0	4.8	0.11	0.218
Ash (g/kg)	101.5	99.1	100.6	2.1	0.731

**Table 5 foods-11-01418-t005:** The effect of dietary stevia and length of consumption on egg * lutein, zeaxanthin, total carotenoids and vitamin E concentration (µg/g).

Time/Diet	Lutein	Zeaxanthin	Total Carotenoids	Vit E
Diet				
C	2.34	0.22	2.61	46.30
C +1% ST	3.14	0.30	3.52	45.99
C+ 2% ST	3.68	0.35	4.15	44.63
SEM	0.089	0.013	0.101	1.125
Time (days)				
0	1.54	0.20	1.75	57.49 ^c^
7	3.94	0.36	4.38	47.04 ^b^
14	2.47	0.23	2.75	43.96 ^b^
21	3.70	0.32	4.16	35.39 ^a^
28	3.60	0.34	4.11	44.33 ^b^
SEM	0.115	0.017	0.131	1.452
Diet × time				
C 0	1.59 ^a^	0.19 ^a^	1.78 ^a^	58.98
C 7	3.08 ^cd^	0.28 ^bcd^	3.38 ^b^	50.69
C 14	1.95 ^ab^	0.18 ^a^	2.14 ^a^	45.30
C 21	2.49 ^bc^	0.21 ^ab^	2.79 ^b^	34.02
C 28	2.58 ^c^	0.24 ^abc^	2.96 ^b^	42.50
C +1% ST 0	1.45 ^a^	0.18 ^a^	1.63 ^a^	55.63
C +1% ST 7	4.19 ^ef^	0.38 ^ef^	4.67 ^cd^	45.20
C +1% ST 14	2.71 ^c^	0.26 ^abc^	3.02 ^b^	45.31
C +1% ST 21	3.62 ^de^	0.31 ^cde^	4.08 ^c^	35.90
C +1% ST 28	3.71 ^e^	0.36 ^def^	4.23 ^c^	47.91
C + 2% ST 0	1.58 ^a^	0.24 ^abc^	1.82 ^a^	57.86
C + 2% ST 7	4.55 ^fg^	0.40 ^f^	5.10 ^de^	45.23
C + 2% ST 14	2.76 ^c^	0.24 ^abc^	3.07 ^b^	41.26
C + 2% ST 21	4.98 ^g^	0.44 ^f^	5.63 ^e^	36.25
C + 2% ST 28	4.52 ^fg^	0.41 ^f^	5.16 ^de^	42.57
SEM	0.199	0.029	0.226	2.515
*p* values				
Diet	<0.001	<0.001	<0.001	0.539
Time	<0.001	<0.001	<0.001	<0.001
Diet × time	<0.001	0.040	<0.001	0.432

* Analyses were performed on the mixed yolk and albumen. Values in a column with different letters differ significantly. C = control diet; SEM = pooled standard error of means; *p* = Fisher’s probability; values in a row with different letters differ significantly.

**Table 6 foods-11-01418-t006:** The effect of dietary stevia and length of storage on egg quality variables.

	Egg Weight (g)	Albumen pH	Albumen Height	Haugh Units	Yolk pH	Yolk Color DSM Yolkfan
Diet						
C	57.55	8.83	5.74	72.87	6.15 ^a^	2.00 ^a^
C + 1% ST	56.38	8.85	5.60	72.07	6.28 ^b^	3.07 ^b^
C + 2% ST	55.24	8.86	5.33	70.27	6.20 ^ab^	3.50 ^c^
SEM	1.155	0.026	0.261	2.026	0.032	0.146
Storage (d)						
0	57.53 ^a^	8.42 ^a^	8.14 ^c^	90.39 ^c^	6.31 ^a^	3.10 ^b^
14	56.14 ^ab^	9.04 ^b^	4.64 ^b^	65.80 ^b^	6.03 ^b^	2.39 ^a^
28	55.50 ^b^	9.08 ^b^	3.89 ^a^	59.02 ^a^	6.29 ^a^	3.08 ^b^
SEM	0.513	0.026	0.146	1.096	0.035	0.090
*p*-values						
Diet	0.386	0.782	0.530	0.657	0.027	<0.001
Storage	0.040	<0.001	<0.001	<0.001	<0.001	<0.001
Diet × storage	0.353	0.391	0.291	0.204	0.405	0.183

C = control diet; SEM = pooled standard error of means; *p* = Fisher’s probability; values in a row with different letters differ significantly.

## Data Availability

The data that support the findings of this study are available from the corresponding author, upon reasonable request, subject to restrictions and conditions.

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
