# Peer review of "Stevia (Stevia rebaudiana) Improves Carotenoid Content in Eggs When Fed to Laying Hens"

_foods, 2022, doi:10.3390/foods11101418_

Round 1

Reviewer 1 Report

This manuscript investigated the effects of Stevia on the production performance, egg quality, egg nutrient composition ,as well as its antioxidant content of laying hens. At the same time, the effects of storage period on egg quality and the antioxidant content of eggs were discussed. The results have guiding significance for the development of feed resources to improve egg quality.

    It is suggested that the manuscript be supplemented or modified as follows:

1 Supplement the ingredient information and source of Stevia.

2.As for the determination of egg and eggshell quality, the number of eggs collected is too small, and there is only one egg in each repetition.

3. As the author mentioned, the storage temperature of eggs in this study is 15 ℃, which is lower than the common storage temperature of eggs. More tests are needed to carry out under  the common storage temperature to clarify the effect of supplementation of stevia on eggs storage period.

Reviewer 2 Report

General comments:

  • The manuscript discusses including phytogenic sources (Stevia) in laying hens feed to achieve health beneficial effects for eggs than the normal ones.
  •  Economic impact of production such types of eggs should be included in either introduction or discussion section.
  • The studied parameters are lacking for any indicators of antioxidants effects on eggs such as total antioxidant capacity or MDA …. etc.
  •  

Specific comments:

  • L75: How feeding antioxidants reduce losses in eggs shell?
  • L93: Please indicate the percentage per Kg not 100 Kg.
  • L94: Please indicate on what basis did the authors choose these dietary experimental levels of stevia. Also, what is the form of used stevia leaves (fresh or dried) and the method of mixing it with diet?
  • L 110:  What is the average egg production at the beginning of the experiment?
  • L182: The analysis in the table is it based on dry matter ?? please clear.
  • L279: This is not clearly appeared in the objectives.

Reviewer 3 Report

Regarding the manuscript entitled'' Feeding stevia (Stevia rebaudiana) to laying hens improves antioxidant content in eggs'' I have some comments that should be considered to improve the manuscript.

The title of the manuscript should be more informative and the authors should choose others stronger keywords.

L17-18. Eye health, I suggest to remove eye and instead human health.

L21. If the feedstuff included in the diet more than 1% it is not supplement it is considered as a dietary substitution. Please rephrase.

General comment in abstract, this section needs improvements, more information about the experimental design, number of laying hens, age of birds, replications, and length of the trial should be added. P-value for the significance also should be added. The conclusion part of the abstract should be focused on the current findings, please revise.

Introduction

L37-38 add ref,

L43. In vitro should be italic

L51-52. Add ref

L72. Please change lacking to very limited

L82. Hypothesis is missing

Materials and Method

L85. Please add the ethical approval number

Why the authors did not include the stevia as a component in the diet why it is supplement the level is more than 1%, this will increase the cost of the diet, and instead we can add carotenoids and lutein in the diets as additives. This is the main flaw-back of the current manuscript. In addition, short experimental period, only 4 weeks

L193. Inclusion?

Table 3. the authors should present feed intake as daily and should calculate egg mass and FCR. Egg production also should be as %. Body weight at the beginning and the end of the experiment should be included.

Egg shell thickness, g please check?? Mm

For tables please add footnote describes the abbreviations in tables.

L206-207. Data should be either in table or figure.

Discussion

The authors ignoring the effect of stevia on the production performance of laying hens.

Round 2

Reviewer 3 Report

Thank you for the revision.